# Which Surgery for Stage II–III Empyema Patients? Observational Single-Center Cohort Study of 719 Consecutive Patients [note 1]

**DOI:** 10.3390/jcm12010136

**Published:** 2022-12-24

**Authors:** Sara Ricciardi, Delia Giovanniello, Francesco Carleo, Marco Di Martino, Massimo O. Jaus, Sara Mantovani, Stefano Treggiari, Luigi Tritapepe, Giuseppe Cardillo

**Affiliations:** 1Unit of Thoracic Surgery, Azienda Ospedaliera San Camillo Forlanini, Carlo Forlanini Hospital, Circonvallazione Gianicolense 87, 00152 Rome, Italy; 2PhD Program, Alma Mater Studiorum, University of Bologna, 40126 Bologna, Italy; 3Department of Cardio-Thoraco-Vascular Surgery, Sapienza University of Rome, Piazzale Aldo Moro 5, 00185 Rome, Italy; 4Departments of Anesthesiology, Critical Care, and Pain Medicine, San Camillo Forlanini Hospital, 00152 Rome, Italy; 5Unicamillus—Saint Camillus University of Health Sciences, 00131 Rome, Italy

**Keywords:** pleural empyema, surgery, minimally invasive surgery, VATS, outcomes

## Abstract

Objective: Recent guidelines support the use of thoracoscopic surgery in stage II-III empyema; however, there is still debate regarding the best surgical approach. The aim of our study is to compare postoperative outcomes of VATS and open surgical approaches for the treatment of post-pneumonic empyema. Methods: Observational cohort study on prospectively collected cases of post-pneumonic empyema surgically treated in a single center (2000–2020). Patients were divided into an open group (OT, posterolateral muscle sparing thoracotomy) and VATS group (VT, 2 or 3 port ± utility incision). The primary outcome of the study was empyema resolution, assessed by the recurrence rate. Secondary outcomes were mortality, complications, pain and return to daily life. All patients were followed up at 1, 3 and 6 months after surgery in the outpatient clinic with a chest radiograph/CT scan. Results: In total, 719 consecutive patients were surgically treated for stage II–III empyema, with 644 belonging to the VT group and 75 to the OT group. All patients had a clinical history of pneumonia lasting no more than 6 months before surgery, and 553 (76.9%) had stage II empyema. Operative time was 92.7 ± 6.8 min for the OT group and 112.2 ± 7.4 for the VT group. The conversion rate was 8.4% (46/545) for stage II and 19.2% (19/99) for stage III. Twelve patients (1.86%) in the VT group and four patients (5.3%) in the OT group underwent additional surgery for bleeding. Postoperative mortality was 1.25% (9/719): 5.3% (4/75) in OT and 0.77% (5/644) in VT. Postoperative stay was 10 ± 6.5 days in OT and 8 ± 2.4 in VT. Overall morbidity was 14.7% (106/719): 21.3% (16/75) in OT and 13.9% (90/644) in VT. In VT, six patients (0.93%) showed recurrent empyema: five were treated with chest drainage and one with additional open surgery. Conclusions: Our findings suggest that the VATS approach, showing a 99% success rate, shorter length of stay and lower postoperative morbidity, should be considered the treatment of choice for thoracic empyema.

## 1. Introduction

It is estimated that approximately 250,000 people with pneumonia died in Europe in 2019 [1].

The occurrence of pleural effusion is associated with a 3–6-fold increase in mortality and complications in up to 50% of patients affected by pneumonia [2].

Most parapneumonic effusions (about 85%) completely resolve with medical therapy and antibiotic treatment. The remaining 15% are characterized by direct bacterial invasion of the pleural cavity, evolving into empyema [3].

According to the American Thoracic Society’s classification, pleural empyema is classified as follows: stage I, uncomplicated pleural effusion without loculation; stage II, fibrinopurulent effusion with loculation, also called complicated pleural effusion; and stage III, pleural thickening with trapped lung [4].

However, in clinical practice, patients rarely present in the “pure” stage, due to the pathological continuum of pleural space infections. This occurrence can further complicate the management of pleural empyema, which remains controversial. Surgical drainage is needed in roughly one third of patients with pleural infection [5] and can be effective in stage I. When drainage alone does not lead to “restitutio ad integrum” and lung re-expansion is not achieved, surgery should be performed: surgical intervention should be recommended for stage II and III, in which organizing fibrinous clots and the development of fibrotic peel could determine lung entrapment [6].

Historically, the surgical treatment of this disease, especially in stage III patients, was open decortication via thoracotomy. However, more recently, several guidelines and robust evidence support the use of VATS for empyema treatment [7,8,9,10,11].

The favorable outcomes of early VATS debridement were stated by two small, randomized studies that demonstrated greater results when minimally invasive surgery was performed, compared to chest tube insertion, in organized empyema [7,12]. Several studies have demonstrated VATS success rates of 82–92% [8,9].

Moreover, VATS showed comparable results in terms of resolution, and superior outcomes in terms of decreased LOS, morbidity, operative time, pain, air leak, duration of chest drainage and early return to daily activities compared to thoracotomy [10,11,13,14,15,16,17,18,19,20]. 

As for the adult population, the use of VATS was recently recommended for pediatric patients, since it leads to the early resolution of infection and a shorter hospital stay [21].

The aim of our study is to evaluate the comparative results of VATS versus an open approach in a series of 719 consecutive patients with post-pneumonic pleural empyema treated in a single center.

## 2. Materials and Methods

Clinical medical records of patients with pleural empyema were reviewed retrospectively. Patients were referred to our department (Unit of Thoracic Surgery, San Camillo Forlanini Hospital, Rome, Italy) from January 2000 to December 2020 after the failure of conservative treatments. All patients had stage II-III post-pneumonic pleural empyema, with a history of pneumonia <6 months before surgery. Exclusion criteria were as follows: patients <18 years, presence of TBC, HIV positivity, fungal infections, immunocompromised states, absence of thoracocentesis/drainage insertion and/or pleural fluid analysis and culture, fibrinolytic use before surgery.

The diagnosis of pleural infection was obtained by imaging (ultrasound and/or computed tomography) and percutaneous aspiration of pleural fluid by thoracentesis or drainage insertion. Empyema stage assessment was performed in all patients based on clinical (duration of symptoms, length of hospital stay) and chest CT characteristics (loculations, idropneumothorax, air-fluid levels, pleural thickening).

Details concerning baseline demographic features (sex, age), clinical presentations, duration of symptoms, treatment received before surgical referral, and comorbidities were collected.

All patients underwent thoracentesis or chest tube insertion and were treated with numerous antibiotics prior to surgery. Patients treated with fibrinolytic agents before surgery were excluded. Antibiotics were selected according to sensitivity results from pleural fluid collection or blood culture analysis. Before microbiological analysis, empirical therapy with piperacillin–tazobactam 4.5 g, three times a day, was usually administered. 

The primary outcome of the study was empyema resolution, assessed by the recurrence rate. Secondary outcomes were mortality, complications, pain and return to daily life.

Length of hospital stay was defined as the number of days from admission to discharge after the surgical procedure. Details on surgery, clinical outcomes, duration of intercostal drainage, complications, duration of intubation and hospitalization were collected and reviewed. The decision to perform an open or a VATS procedure was made by the surgeon according to his/her preferences and expertise. 

Patients were divided into two groups: those treated by thoracotomy (OT group) and those undergoing VATS (VT group). Following the principles of intent to treat analysis, in the case of conversion from VATS to open surgery, patients were not considered in the OT group. A subset analysis was performed in stage III empyema to compare VT and OT groups.

Compelling postoperative pain control was accomplished with a standardized protocol for pain management. Opioid-based intravenous patient-controlled analgesia (continuous infusion of 2 mg morphine + 60 mg ketorolac + 2 mg ondansetron every 24 h) was administered from day 0 to day 4. Acetaminophen 1 g was dispensed three times a day (h 8:00–16:00–24:00) from day 0 until the chest tubes were removed. Ketorolac was used as needed (30 mg, maximum twice a day). Our postoperative pain management protocol was modified in December 2020; hence, all patients in the present dataset were treated with the same approach. Postoperative pain was evaluated using a 10-point numeric scale questionnaire (1 being no pain; 10 worst pain ever experienced), recorded on day 1 and 6 after surgery, and after 6 months during an outpatient visit. Respiratory physiotherapy, including positive airway pressure inhalation and mobilization as early as feasible, was conducted from postoperative day 1.

Lung re-expansion was postoperatively assessed by chest X-ray at day 0 and postoperative day 1, 3 and 7, unless postoperative complications occurred. Before discharge, all patients underwent chest CT to evaluate the lung and pleural space (Figure 1 and Figure 2).

All patients were followed up at 1, 3 and 6 months after surgery with a clinical interview and chest radiograph/CT scan.

All patients provided informed consent to surgery and the inclusion of their clinical information in our database. This study was a retrospective analysis of standard surgical procedures and was conducted in accordance with the Declaration of Helsinki and approved by our internal institutional review board (IRB).

### 2.1. Surgical Technique

All patients underwent general anesthesia with a double lumen endotracheal tube and were placed in a lateral position, with the ipsilateral lung deflated.OT group

Postero-lateral serratus anterior-sparing thoracotomy with rib spreading (no rib resection) was performed. Complete and thorough inspection, debridement of the whole pleural surface and decortication (including visceral and parietal pleura) were performed to achieve complete lung expansion, including the fissures.

At the end of the procedure, two large-bore (30 CH and 32 CH) chest tubes were placed through separate incisions.VT group

Three-port or two-port procedures were performed. The more suitable intercostal spaces were selected according to the preoperative CT scan. The area of pleural collection was detected by aspiration of the fluid with a needle, and then the first 1-cm port was placed.

3-port: Once a pleural space was created, the remaining two ports (1 cm each) were placed under thoracoscopic vision to avoid injury to the underlying lung parenchyma. No utility incision was performed. No retractor was used.

2-port: Utility incision was performed under thoracoscopic vision at IV or V intercostal space. The Alexis XS retractor was used.

Under thoracoscopic vision, complete lung decortication (when needed), fluid evacuation, loculations and septa removal by use of a sucker, ring clamp, peanut dissector and endoscopic forceps were achieved. Material for microbiological analysis was collected in all patients. Achieving decortication, adherent thickened pleura was cautiously removed and the lung was entirely freed from the apex to the diaphragm. All empyematic collections were removed and parietal pleura was detached (Figure 3 and Figure 4). If complete deloculation and decortication could not be achieved, conversion to an open procedure was performed. At the end of the procedure, two chest tubes (30 CH and 32 CH) were placed.

Minimal lung parenchymal injury is mandatory to prevent air leak. Repair of all suspected lung damages and careful hemostasis was also achieved with the application of fibrin glue (Tisseel^®^, Baxter, Deerfield, MA, USA) and a Sealing Hemostat (TACHOSIL^®^, Takeda Pharmaceutical, Tokyo, Japan; Haemopatch^®^ Baxter, Deerfield, MA, USA) either in open or in VATS procedures. 

### 2.2. Statistical Analysis

Baseline variables were described with percentages for categorical variables and mean and standard deviation (SD) for continuous variables. Normality of continuous variables was studied with the Kolmogorov–Smirnov test.

Statistical analysis of surgical outcomes (operative time, redo surgery, hospital stay, postoperative pain on day 1 and day 6 and at 6 months after surgery, complications, postoperative air leak and time to return to work) was performed in the OT group and VT group. An unpaired *t*-test and Mann–Whitney U test were used when indicated. All tests were two-tailed. We considered values of *p* < 0.05 as statistically significant. Statistical analysis was conducted using SPSS Statistics software (IBM SPSS Statistics 20, IBM Corporation, Chicago, IL, USA).

## 3. Results

Among 719 patients, 644 (89.56%) were treated with minimally invasive surgery (VT group) and 75 (10.46%) with an open approach (OT group). In the VT group, the conversion rate was 8.4% (46/545) in stage II empyema (delay between hospitalization and surgery < 4 weeks) and 19.2% (19/99) for stage III (delay between hospitalization and surgery 4–6 weeks) (*p* = 0.001). The average ASA grade was 2.9. ASA grades 1, 2, 3 and 4 were reported in 24, 127, 529 and 39 cases, respectively. Severe comorbidities were common among patients, with a prevalence of cardiac disorders (Table 1).

Postoperatively, 571 patients required admission to the Intensive Care Unit (ICU), while the remaining 148 patients returned to the surgical ward. The median length of ICU stay was 3 days (1–10 days). Details on postoperative outcomes are reported in Table 2.

The overall postoperative mortality was 1.25% (9/719 cases). Four patients died in the OT group, which resulted in a 5.3% (4/75) mortality rate: one patient died of sepsis 41 days after the operation and three patients died of unrelated disease (one of bowel perforation, one of peptic ulcer, one of ictus cerebri and one of myocardial infarction, on the 15th, 18th and 20th postoperative day, respectively). Five patients died in the VT group, with a 0.77% (5/644) mortality rate: one patient died of sepsis 50 days after surgery and four patients died of unrelated disease (one of peptic ulcer, two of myocardial infarction, one of ictus cerebri, on the 25th, 15th, 17th and 11th postoperative day, respectively). Reoperation was performed in sixteen patients due to bleeding: twelve in the VT group (1.86%) and four in the OT group (5.3%).

Postoperative overall morbidity was 14.7% (106/719) and 21.3% (16/75) in OT and 13.9% (90/644) in VT, respectively. The most common complications were prolonged air leak (*n* = 60), atrial fibrillation (*n* = 22), renal insufficiency (*n* = 12), residual pleural space (*n* = 8), wound infection or dehiscence of open thoracotomy (*n* = 4).

Six patients (0.93%, all in the VT group) showed recurrent empyema: five were successfully treated with chest drainage, and one with additional open surgery on the 32nd postoperative day.

The reported pain on POD 1 and 6 was significantly lower for patients in the VT group (*p* < 0.0001); however, no significant differences were reported in terms of chronic pain (*p* = 0.08).

A significantly reduced postoperative length of stay was reported in the VT group, with 8 ± 2.4 days, compared to the OT group, with 10 ± 6.5 days (*p* < 0.0001). Similarly, the time to return to work was lower in VT, with 23.4 ± 4.8 days, versus 33.2 ± 8.6 in OT (*p* < 0.0001).

A subset analysis was conducted on 166 patients with stage III empyema. Overall, 67 patients were treated with the open approach and 99 with VATS. All patients required debridement and extended visceral and parietal pleural decortication. Stage III patients in the VT group showed equivalent results to the OT group in terms of empyema resolution and superior results in terms of air leak (<0.0001), length of stay (<0.0001), acute pain (<0.0001) and return to work (<0.0001).

## 4. Discussion

Empyema presentation can be delayed due to subacute onset, with a reported median LOS of 14–19 days [3,14,22]. If early-stage empyema can be promptly treated and solved with chest tube drainage, antibiotics and fibrinolytic drugs, as stated by the British Thoracic Society (BTS) guidelines, chronic empyema is rather associated with significant healthcare resource consumption, intensive and prolonged use of antibiotics, chest tube drainage and surgery [6].

Nowadays, there is no consensus on the best surgical option for empyema treatment. However, several studies have reported superior postoperative outcomes for the VATS approach versus open thoracotomy, with comparable results in terms of resolution [7,8,9,10,11].

The primary outcome of the present study was empyema resolution. Lung re-expansion was achieved in 100% of OT and 99.8% of VT patients. The recurrence rate of the whole cohort was 0.83%: six patients (0.93%, all belonging to the VT group, versus 0% in the OT group) showed recurrent empyema. Notably, all these patients had stage III empyema, with a delay to surgery > 6 weeks. Stage III empyema was also a risk factor for conversion: we reported a conversion rate of 8.4% (46/545) in stage II empyema (delay between hospitalization and surgery < 4 weeks) and 19.2% (19/99) in stage III (delay between hospitalization and surgery 4–6 weeks) (*p* = 0.001). Early referral to surgery was suggested to facilitate the possibility of treating patients with VATS and reducing the likelihood of conversion [23]. Several studies have shown that the time to referral is the most important independent factor for conversion [20,24,25]. Moreover, data across the large STS database reported adverse outcomes in terms of readmission, major morbidity, prolonged LOS and discharge to care other than home, being higher when preoperative hospitalization was longer than 5 days [13].

Secondary outcomes were mortality, complications, pain and return to daily life.

In our series, an extremely low postoperative mortality rate was reported in the VT group (0.77%), which was significantly lower compared to the OT group (5.3%) (*p* = 0.008). Even if patients included in the OT group mostly died due to unrelated disease, given that the VT and OT groups had a comparable percentage of comorbidities, the VATS approach showed a safer profile compared to the open approach. 

The VATS approach showed statistically significant superior outcomes in terms of acute postoperative pain (*p* < 0.0001), postoperative air leak (*p* < 0.0001), LOS (*p* < 0.0001) and time to return to work (*p* <0.0001). The redo surgery rate for bleeding was higher in the OT group, at 5.3% (4/75), than the VT group, at 1.86% (12/644) (*p* = 0.05). All patients had stage III empyema. The reported morbidity rate was lower in the VT group, at 13.9% (90/644), compared to the OT group, at 21.3% (16/75).

Our findings were in line with the literature [16,17,18,19,20,21,22,23,24,25,26,27].

Moreover, the largest patient cohort existing so far in the General Thoracic Surgery Database (Society of Thoracic Surgeons), accounting for over 7300 patients undergoing decortication, showed a statistically significant difference in mortality between VATS (2.8%) and thoracotomy (3.7%) with open surgery, which was also related to increased morbidity, prolonged hospitalization and discharge to care other than home [13]. The American Association for Thoracic Surgery (AATS) expert consensus guidelines for the management of empyema suggest the use of VATS for stage II and III, highlighting a general shift among practicing surgeons away from open thoracotomy to VATS [17].

## 5. Limitations

The most important limitation of the present study is its retrospective nature, with a wide timeframe and a non-homogeneous number of patients in each group. Moreover, thoracotomy and VATS rates were not equally distributed throughout the study period, which lasted over twenty years. During the last ten years, in our center, the great majority of surgical procedures were performed by VATS. This could be considered a confounding factor, even if it is attributable to the evolution of the technique and to the results of the present prospective cohort of patients.

Nevertheless, this series represents one of the largest cohorts of consecutive patients treated in a single center. Our results, in line with the literature, support the use of minimally invasive surgery in stage II-III empyema patients

## 6. Conclusions

In conclusion, according to the great majority of recent studies [28], in the present series, in the absence of randomized clinical trials, the VATS approach seemed to improve patients’ outcomes and can be considered a fitting treatment for stage II and III pleural empyema. However, if complete decortication and deloculation cannot be achieved by VATS, an open procedure is recommended.

We highlight the importance of the prompt referral of empyema patients to a thoracic surgical unit for evaluation and rapid treatment. Late referral can lead to a more prolonged clinical course, with high conversion rates and worse postoperative outcomes.

## Figures and Tables

**Figure 1 jcm-12-00136-f001:**
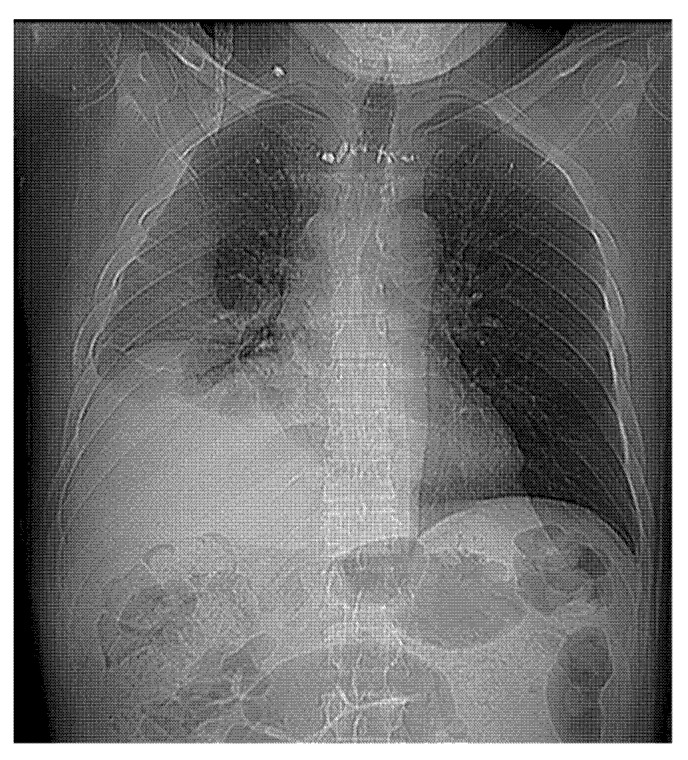
Preoperative CT scan.

**Figure 2 jcm-12-00136-f002:**
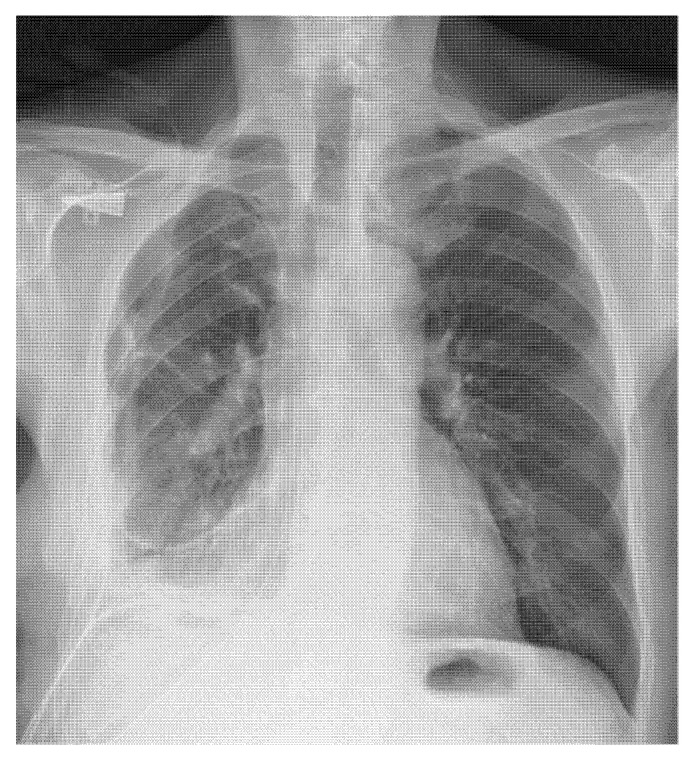
Postoperative X-ray.

**Figure 3 jcm-12-00136-f003:**
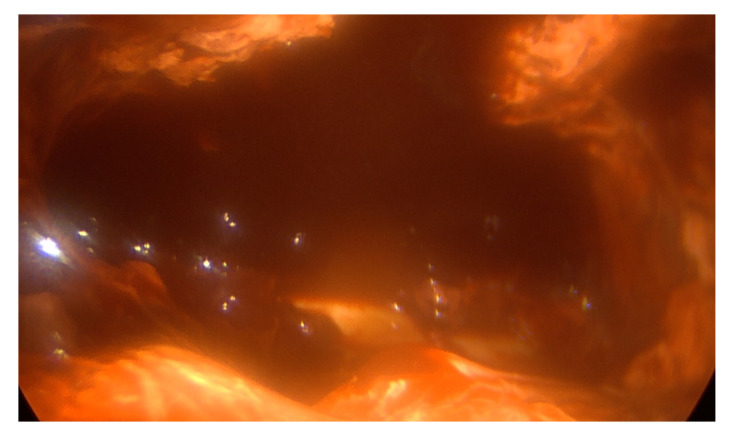
Intraoperative VATS view.

**Figure 4 jcm-12-00136-f004:**
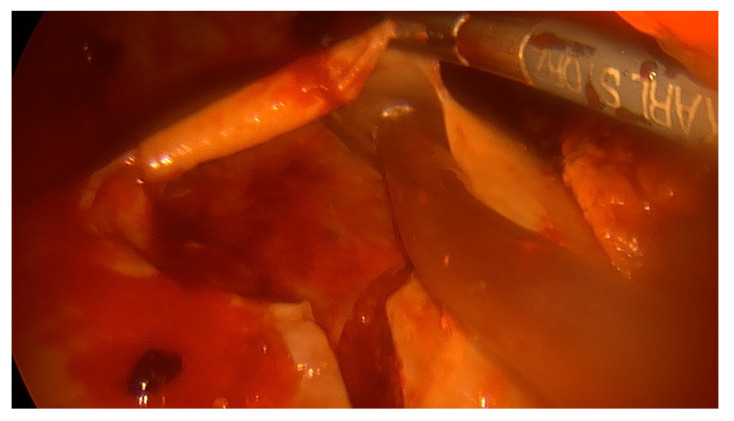
VATS decortication.

**Table 1 jcm-12-00136-t001:** Preoperative patients’ characteristics.

Characteristics	OPEN (*n* = 75)	VATS (*n* = 644)	*p* Value
Male/female	43/32	471/173	
Age (range)	58 (23–83)	56 (17–82)	
Symptoms duration (days)	7 ± 2	5 ± 2	
Preoperative treatment:			
chest drainage	63 (84%)	543 (84.3%)	0.473
thoracentesis	12 (16%)	101 (15.7%)	0.475
Smoking history	25 (33.3%)	141 (21.9%)	0.013
Intravenous drug abuse	11 (14.6%)	23 (3.5%)	0.000
Comorbidities			
Diabetes	51 (68%)	432 (67%)	0.430
Alcoholism	24 (32%)	81 (12.6%)	0.000
Cardiovascular	45 (60%)	376 (58.4)	0.395
Liver cirrhosis	18 (24%)	26 (4%)	0.036
Empyema stage II	8 (10.6%)	545 (84.6%)	
Empyema stage III	67 (89.3%)	99 (15.4%)	

**Table 2 jcm-12-00136-t002:** Postoperative outcomes.

Paramether	OPEN (*n* = 75)	VATS (*n* = 644)	*p* Value
Operative time (min, SD)	92.7 ± 6.8	112.2 ± 7.4	<0.0001
Re-expansion of the lung (*n*, percentage)	75 (100%)	643 (99.8%)	n.s.
Redo surgery (bleeding)	5.3% (4/75)	1.86% (12/644)	0.05
Postoperative mortality	5.3% (4/75)	0.77% (5/644)	0.008
Morbidity	21.3% (16/75)	13.9% (90/644)	0.08
Postoperative air leak (days)	4.3 ± 3,8	3.1 ± 2.6	<0.0001
Postoperative lenght of stay (days)	10 ± 6.5	8 ± 2.4	<0.0001
Postoperative pain (median, range)			
1 and 6 days after surgery	6 (4–9)	4 (3–6)	<0.0001
6 months	2 (0–4)	2 (0–3)	0.08
Time to return to work (days, SD)	33.2 ± 8.6	23.4 ± 4.8	<0.0001

## Data Availability

The data underlying this article will be shared on reasonable request to the corresponding author.

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
