# Peer review of "Which Surgery for Stage II–III Empyema Patients? Observational Single-Center Cohort Study of 719 Consecutive Patients†"

_jcm, 2022, doi:10.3390/jcm12010136_

Round 1

Reviewer 1 Report

In their single-center observational cohort study, the authors compared the outcomes of VATS and open surgical approaches for the treatment of postpneumonic empyema in 719 patients. They concluded that the VATS approach should be considered the treatment of choice for thoracic empyema, with a 99% success rate, shorter length of stay, and lower postoperative morbidity.

I read the study with interest. Although this study contains no new conclusions, I believe it should be considered for publication because of the large sample size and long-term follow-up. My concerns are as follows:

1. Abstract - Please indicate outcomes of the study under methodology

2. In the introduction, the authors should emphasize the benefits of VATS for pleural empyema and why it is important to perform VATS before the stage III occurs.

3. In the introduction, the authors rightly emphasize that multiple guidelines and robust evidence support the use of VATS in the treatment of empyema. They have only provided references for adults, but this also applies for pediatric patients. This should also be highlighted and supported with reference (REF: Pogorelić Z, Bjelanović D, Gudelj R, Jukić M, Petrić J, Furlan D. Video-assisted thoracic surgery in early stage of pediatric pleural empyema improves outcome. Thorac Cardiovasc Surg. 2021;69(5):475-480. doi: 10.1055/s-0040-1708475.)

4. Methodology-The authors indicated that details of demographic characteristics, clinical presentation, duration of symptoms, treatment received prior to surgical referral, and comorbidities were recorded. For each subset of the data, please provide the exact parameters that were collected. E.g., demographic data (age, sex, BMI, race...)

5. The authors stated that antibiotics were selected according to the results of sensitivity testing of pleural sampling or blood culture. I assume that the patients were already receiving antibiotic therapy before the microbiological analysis. The authors should indicate their initial treatment protocol, which antibiotics they used and at what dosage before receiving the results of the microbiological analysis.

6. What were the criteria for performing open surgery or VATS, depending on surgeon preference and skill or? Please specify.

7. Opioid-based intravenous patient-controlled analgesia - Please specify which drug and at what dose. Also provide a dose for acetaminophen. Please use the generic name for 'ketorolac' and also provide the dosage.

8. The authors stated that the study was approved by their internal institutional review board (IRB). Please provide the IRB reference and date of approval.

9. What statistical test was used to test normality of distribution of the data.

10. Table 2 - Please indicate the significance of the values in each row. E.g., operative time (min; mean ± SD) or morbidity, n (%). In addition, all abbreviations should be mentioned in a table legend. Decimal numbers should be represented with a comma (not a period).

11. The discussion should be improved. The discussion section needs to be revised/reorganized. Do not provide a review of the literature in this section. Do not discuss your findings piecemeal. Focus on the results of the main objectives of the study. Write in four consecutive paragraphs (without headings): (i) summary (not data) of the findings of this study; (ii) logical and coherent comparison with the existing literature, focusing the comparison on the main objective(s); (iii) limitations of the study; and (iv) implications for practice/policy/research with a concluding statement.12. Limitations of the study should be listed at the end of the discussion rather than after the conclusions.

13. The manuscript should be proofread by an English editor or native speaker.

14. Please consider including some intraoperative photographs, if available.

Author Response

Dear reviewer,

we would like to thank you for your comments and suggestion. The manuscript has been modified according to your advices. 

  1. Abstract - Please indicate outcomes of the study under methodology. ADDED
  2. In the introduction, the authors should emphasize the benefits of VATS for pleural empyema and why it is important to perform VATS before the stage III occurs. added, lines 63-70
  3. In the introduction, the authors rightly emphasize that multiple guidelines and robust evidence support the use of VATS in the treatment of empyema. They have only provided references for adults, but this also applies for pediatric patients. This should also be highlighted and supported with reference (REF: Pogorelić Z, Bjelanović D, Gudelj R, Jukić M, Petrić J, Furlan D. Video-assisted thoracic surgery in early stage of pediatric pleural empyema improves outcome. Thorac Cardiovasc Surg. 2021;69(5):475-480. doi: 10.1055/s-0040-1708475.) added, lines 71-72
  4. Methodology-The authors indicated that details of demographic characteristics, clinical presentation, duration of symptoms, treatment received prior to surgical referral, and comorbidities were recorded. For each subset of the data, please provide the exact parameters that were collected. E.g., demographic data (age, sex, BMI, race...) added, line 88
  5. The authors stated that antibiotics were selected according to the results of sensitivity testing of pleural sampling or blood culture. I assume that the patients were already receiving antibiotic therapy before the microbiological analysis. The authors should indicate their initial treatment protocol, which antibiotics they used and at what dosage before receiving the results of the microbiological analysis. added, lines 96-98
  6. What were the criteria for performing open surgery or VATS, depending on surgeon preference and skill or? Please specify. Added lines 104-5
  7. Opioid-based intravenous patient-controlled analgesia - Please specify which drug and at what dose. Also provide a dose for acetaminophen. Please use the generic name for 'ketorolac' and also provide the dosage. Added, lines 109-113
  8. The authors stated that the study was approved by their internal institutional review board (IRB). Please provide the IRB reference and date of approval. Updated, not necessary as local law on retrospective analysis on daily practice
  9. What statistical test was used to test normality of distribution of the data. Kolmogorov–Smirnov test. Added lines 175-6
  10. Table 2 - Please indicate the significance of the values in each row. E.g., operative time (min; mean ± SD) or morbidity, n (%). In addition, all abbreviations should be mentioned in a table legend. Decimal numbers should be represented with a comma (not a period). done
  11. The discussion should be improved. The discussion section needs to be revised/reorganized. Do not provide a review of the literature in this section. Do not discuss your findings piecemeal. Focus on the results of the main objectives of the study. Write in four consecutive paragraphs (without headings): (i) summary (not data) of the findings of this study; (ii) logical and coherent comparison with the existing literature, focusing the comparison on the main objective(s); (iii) limitations of the study; and (iv) implications for practice/policy/research with a concluding statement.12. Limitations of the study should be listed at the end of the discussion rather than after the conclusions. Edited and reorganized
  12. The manuscript should be proofread by an English editor or native speaker. Done
  13. Please consider including some intraoperative photographs, if available. Added figure 3 and 4

Reviewer 2 Report

Dear Editor and Authors,

Thank you for asking me to evaluate this review manuscript titled “Which Surgery for Stage II-III Empyema Patients? Observational Single Centre Cohort Study on 719 Consecutive Patients” by Dr. Ricciardi and colleagues from the Unit of Thoracic Surgery at Azienda Ospedaliera San Camillo-Forlanini in Rome, Italy.

In this single institution, retrospective case control study the authors compare two different surgical approaches thoracoscopic (VATS) versus open thoracotomy for drainage and decortication of complex (stage II – III) empyema. They conducted an analysis and comparison of outcome in 719 patients and concluded that thoracoscopy/VATS has a significant success rate approaching 99%, has a shorter length of stay and lower postoperative morbidity and as such it should be considered as the treatment of choice for thoracic empyema. This is a relatively well conducted study with given its inherent limitations has a good methodology and clear primary and secondary end points.

I do have a few comments and suggestions to make regarding this work:

Comments:

1.       The manuscript is well written and overall easy to understand but some minor English language corrections are needed especially regarding spelling and expression mistakes. I suggest a “polishing up” by a native language speaker or a professional service!

2.       The submitted post-operative chest X-ray/CT Scan (Figures 1 & 2) do not provide anything of value to the manuscript and should be removed.

3.       The authors note that ”If complete deloculation and decortication could not be achieved, conversion to open procedure was performed” however what was their measure of “complete decortication” and how complete was decortication performed by VATS?

4.       What percentage of VATS patients required extensive decortication and which only minor or partial. This is a big issue of bias because, given the retrospective nature or the study, it is quite probable that more chronic, more complex and more extensive cases underwent open thoracotomy instead of VATS!

5.       Why was statistically corrective analysis techniques such as propensity score matching not used to adjust for potential biases (as the one mentioned above for example!!).

6.       Why was not a sample size calculation or power analysis performed prior to data mining to confirm the number of patients analyzed is large enough to produce statistically meaningful outcomes??

7.       The length of the study period (20 years) is quite long and this, although it provides for a large number of patients (see statement above) is a problem onto itself because techniques have changed and evolved, antibiotic management has evolved, anesthetic techniques have changed, ect ect. This should be mentioned as a limitation and discussed in the discussion section.

8.       Why aren’t statistical differences and p-values reported in table 1 for example? It is clear from the data that the open thoracotomy (OT) group had a higher percentage of co-morbidities and high risk patients!

9.       What was the main microorganism isolated (and which other ones were isolated as well - a separate table should be provided or the variable added on table 1) and what was the distribution between the two groups. Is there a difference? Some microorganisms produce more severe disease and extend of empyema than others. This needs to be clarified/analyzed!

10.   Why was such a high proportion of patients admitted to the ICU post procedure? This is not standard unless there is a policy (in some departments there is) for elective prolonged ventilation to facilitate lung re-expansion via positive airway pressures. Is this something that applies here?

11.   The operative mortality reported for the OT is indicative of the higher risk patients in this group as is the fact that most deaths were unelated to the empyema but were a result of other, co-existing pathologies!

12.   In the results section no p-values are reported so it is unclear if the differences were significant in regards to mortality, re-operation, air leak, atrial fibrillation, renal insufficiency, space, recurrent empyema ect ect (all variables reported in lines 182 – 187).

13.   Given that this was a retrospective chart review how complete was the data collection and the data sets? I am interested to find how variables like days until return to work were recorded and assessed. In addition, why was pain assessed peri-operative and long term and how were these variables recorded?

14.   The baseline, surgical characteristics and outcomes should be reported and provided in additional tables which could be part of supplemental data!

15.   The statement made in the discussion section (lines 225 - 227) regarding mortality needs to be quantified or even removed given that mortality in the OT group were unrelated to the surgery but a product of other factors!! This statement therefore is not supported by the data.

16.   Recurrence of empyema in the VATS group as reported in the results section and in the discussion is not a minor drawback but rather it can be attributed (as mentioned previously) of not providing a complete and adequate drainage and decortication via VATS!! This needs to be considered by the authors and it’s a major major limitation of their study and hypothesis that needs addressing in the discussion and in the limitations section.

17.   One can argue that the judgement regarding VATS been the “treatment of choice” as mentioned in the conclusion section is still out!! The data presented herewith by the authors surely are not that conclusive. I suggest toning down the statement given that both procedures most likely have their place in the armamentarium of the thoracic surgeon and it is up to him to choose and perform the most appropriate procedure to the needs of the patient. This is the point the authors should try to make, a uniting/compromising one versus a dividing/absolute one: “VATS is better (why we would like it to be)!!”   

Minor comments:

1.       Line 67: with instead of wit   

In conclusion, this manuscript needs some significant work and corrections to bring it up to standard for publication. There are a number of issues (as mentioned above) the authors need to address and correct but nothing that it is not doable because it does have as previously mentioned a good methodology and structure. Thank you for asking me to review this work and I wish all well. I am awaiting for the revised version which I hope, no I know that if the authors implement the reviewers suggestions will be much improved.

Author Response

Dear reviewer,

we would like to thank you for your advices and comments. The text has been revised according to your suggestion.

Thanks once again for your time and effort.

The manuscript is well written and overall easy to understand but some minor English language corrections are needed especially regarding spelling and expression mistakes. I suggest a “polishing up” by a native language speaker or a professional service! Done

  1. The submitted post-operative chest X-ray/CT Scan (Figures 1 & 2) do not provide anything of value to the manuscript and should be removed. As other reviewer ask for more pictures (intra, pre and postoperative) we have added those pictures.

  1. The authors note that ”If complete deloculation and decortication could not be achieved, conversion to open procedure was performed” however what was their measure of “complete decortication” and how complete was decortication performed by VATS? All lung should be decorticated. The main measure of complete decortication is the fully expansion of the lung. If the complete expansion could not be achieved by VATS a conversion is performed.

  1. What percentage of VATS patients required extensive decortication and which only minor or partial. This is a big issue of bias because, given the retrospective nature or the study, it is quite probable that more chronic, more complex and more extensive cases underwent open thoracotomy instead of VATS! All stage III patients required extensive decortication, this is the reason why a subset analysis was conducted on stage III empyema to easily compare VATS and open approach. .

  1. Why was statistically corrective analysis techniques such as propensity score matching not used to adjust for potential biases (as the one mentioned above for example!!). Our statistician didn’t require it. Just a subset analysis was considered necessary.

  1. Why was not a sample size calculation or power analysis performed prior to data mining to confirm the number of patients analyzed is large enough to produce statistically meaningful outcomes?? This is a retrospective cohort analysis on a large series, statistical analysis was performed on previously collected data. Even in this case the statistician didn’t ask for further analysis.

  1. The length of the study period (20 years) is quite long and this, although it provides for a large number of patients (see statement above) is a problem onto itself because techniques have changed and evolved, antibiotic management has evolved, anesthetic techniques have changed, ect ect. This should be mentioned as a limitation and discussed in the discussion section. Already mentioned in Limitation section, lines 273-279

  1. Why aren’t statistical differences and p-values reported in table 1 for example? It is clear from the data that the open thoracotomy (OT) group had a higher percentage of co-morbidities and high risk patients! Added in table 1

  1. What was the main microorganism isolated (and which other ones were isolated as well - a separate table should be provided or the variable added on table 1) and what was the distribution between the two groups. Is there a difference? Some microorganisms produce more severe disease and extend of empyema than others. This needs to be clarified/analyzed! Great majority of patients had MRSA, Staphylococcus aureus or Pseudomonas, while mixed microorganisms were isolated. It is difficult to conduct an accurate analysis on this data because our previous database didn’t contain this variable.

  1. Why was such a high proportion of patients admitted to the ICU post procedure? This is not standard unless there is a policy (in some departments there is) for elective prolonged ventilation to facilitate lung re-expansion via positive airway pressures. Is this something that applies here? As hospital policy, all patients with ASA 3/4 who belong to departments without sub-intensive care unit, are transferred to ICU after surgery. However, it does not implicate the need of prolonged ventilation.

  1. The operative mortality reported for the OT is indicative of the higher risk patients in this group as is the fact that most deaths were unelated to the empyema but were a result of other, co-existing pathologies! Added lines 254-6

  1. In the results section no p-values are reported so it is unclear if the differences were significant in regards to mortality, re-operation, air leak, atrial fibrillation, renal insufficiency, space, recurrent empyema ect ect (all variables reported in lines 182 – 187). All p value are reported in table 2 and are not added in the result section just to avoid redundancy.

  1. Given that this was a retrospective chart review how complete was the data collection and the data sets? I am interested to find how variables like days until return to work were recorded and assessed. In addition, why was pain assessed peri-operative and long term and how were these variables recorded? All data was recorded by outpatient visit or phone interview. This study follows a previous one conducted in our department and published in 2009.

  1. The baseline, surgical characteristics and outcomes should be reported and provided in additional tables which could be part of supplemental data! We have added those data in text tables

  1. The statement made in the discussion section (lines 225 - 227) regarding mortality needs to be quantified or even removed given that mortality in the OT group were unrelated to the surgery but a product of other factors!! This statement therefore is not supported by the data. Added lines 254-6

  1. Recurrence of empyema in the VATS group as reported in the results section and in the discussion is not a minor drawback but rather it can be attributed (as mentioned previously) of not providing a complete and adequate drainage and decortication via VATS!! This needs to be considered by the authors and it’s a major major limitation of their study and hypothesis that needs addressing in the discussion and in the limitations section. Empyema recurrence was found in only 6/644 patients. We think we can consider it as technique learning curve instead of a major limitation.

  1. One can argue that the judgement regarding VATS been the “treatment of choice” as mentioned in the conclusion section is still out!! The data presented herewith by the authors surely are not that conclusive. I suggest toning down the statement given that both procedures most likely have their place in the armamentarium of the thoracic surgeon and it is up to him to choose and perform the most appropriate procedure to the needs of the patient. This is the point the authors should try to make, a uniting/compromising one versus a dividing/absolute one: “VATS is better (why we would like it to be)!!”    Added lines 285-288

Round 2

Reviewer 1 Report

The authors performed requested corrections. The manuscript has been improved and can be published in current form.

Reviewer 2 Report

Dear Editor and Authors,

So, I have re-reviewed this manuscript again and have considered the responses the authors gave to my comments. Although to be honest a few of my comments especially in regards to statistical issues (i.e. propensity score matching, multivariate modeling ect) were not answered adequately and instead I feel I was given something along the subterfuge "our statistician did not require it, did not do it, we did not ask it ect", nevertheless the manuscript is improved and so I can recommend its publication now. 

Kind regards to all.